# Driving under viral impairment: Linking acute SARS-CoV-2 infections to elevated car crash risks

**Baran Erdik** [1,2]*

**1** Department of Healthcare Administration, American Vision University, Anaheim, California, United States of America, **2** Hygia Health, Miami, Florida, United States of America

* baran@erdik.co.nz

## Abstract

This study explores the linkage between acute SARS-CoV-2 and car crashes across U.S. states, correlating with COVID-19 mitigation strategies, vaccination rates, and Long COVID prevalence. This investigation analyzed aggregate COVID-19 and car crash data spanning 2020–2023, with data collection occurring between March and May 2024. Analysis was done via a Poisson regression model, adjusted for population. Key variables included vaccination status, month-specific effects relating to initial pandemic shutdowns, and Long COVID rates. Results demonstrated a significant association between acute COVID-19 infections and an increase in car crashes, independent of Long COVID status to the tune of an OR of 1.25 [1.23-1.26]. This association was observed despite varying mitigation efforts and vaccination rates across states. The study found no protective effect of vaccination against car crashes, challenging prior assumptions about the benefits of vaccination. Notably, the risk associated with COVID-19 was found to be analogous to driving impairments seen with alcohol consumption at legal limits. Findings suggest significant implications for public health policies, especially in assessing the readiness of individuals recovering from COVID-19 to engage in high-risk activities such as pilots or nuclear plant employees. Further research is necessary to establish causation and explore the exact effects of COVID-19 within the CNS affecting cognition and behavior.

## Introduction

Coronavirus disease 2019 (COVID-19) caused by severe acute respiratory syndrome coronavirus 2 (SARS-CoV-2) first reported in December 2019. Since then, it has resulted in over one million deaths in the US alone as well as an estimated incidence of almost 3.5 infections per person as of early 2024 [1]. Beyond acute hospitalizations and associated mortality and morbidity, one of the main concerns surrounding COVID-19 is the post-COVID-19 conditions termed long COVID or post-acute sequelae of SARS-CoV-2 infection (PASC) which affect at least a quarter of patients or more surviving even mild cases of COVID. A recent longitudinal follow-up study demonstrated that almost 63% of those who have had COVID-19 met the criteria for long-term COVID [2,3].

**Data availability statement:** All relevant data are within the paper and its Supporting information files.

**Funding:** The author received no specific funding for this work.

**Competing interests:** The author has declared that no competing interests exist.

Studies in this context have demonstrated that Long Covid is associated with numerous symptoms affecting all bodily systems and can be clustered within categories. One of the main symptom clusters is cognitive complaints, particularly memory and concentration deficits with data indicating more than 30% of those that have had acute COVID-19 reporting such symptoms [4]. Further, it has been shown that reinfections confer increased risk of Long Covid, and the data demonstrating protective effect of vaccination on Long Covid is conflicting [5,6]. In light of such data, as well as removal of all mitigation efforts in the US, the question of Long Covid evolves into a matter of when, not if [7,8].

Several hypotheses have been proposed regarding Long Covid, mainly viral persistence, immune dysregulation, and viral reactivation [9]. From a neuropathological perspective, it has been shown that immune dysregulation within the immune system is evident with persistent immune activation in the context of antibody secreting B cells [10], microglial activation associated with poor cognitive function [11,12], as well as effects of SARS-CoV-2 on the microvasculature, which can explain pan-cerebral issues along with perturbations of synaptic function in general [13,14]. Further, it's also been shown that macrophages can enter the brain tissue post COVID-19, and stay in perpetuity leading to neural inflammation [15,16]. Not surprisingly, studies have also demonstrated loss of brain volume, increased risk of Alzheimer's and numerous autonomic nervous system issues as postural tachycardia syndrome (POTS), some with sufficient severity that it may lead to Ondine's curse [17–19]. Additionally, neural viral reservoirs may perpetuate immune dysregulation, culminating in T-cell exhaustion [20,21]. Regardless of the pathogenesis, the phenomenon commonly referred to as "brain fog," characterized primarily by cognitive impairments, is becoming increasingly prevalent. Data indicate that even mild acute infections can have lasting effects on brain function, potentially leading to significant challenges in tasks that demand high levels of attention and cognitive processing [22].

One such cognitively demanding task is driving, which relies heavily on psychomotor speed, executive function, and visual processing [23,24]. It is well established that impairments in these domains, as seen in conditions such as ADHD and dementia, correlate with an increased risk of motor vehicle crashes. Despite this, most states do not require specific medical evaluations or tests to obtain a driver's license, with a few exceptions for conditions like epilepsy or for older adults [25–27]. Given the effects of Long COVID and the sheer scale of numbers affected, the full impact of COVID-19 on driving remains largely unknown. In this sense, studies have analyzed increases in car crashes as well as crash mortality in the US contemporarily with COVID-19 pandemic [28] While these analyses did not directly attribute these increases to COVID-19 beyond government-imposed "stay-at-home" orders -shutdowns-, they may have correctly identified a trend without recognizing the role of SARS-CoV-2 as a contributing factor. The virus has been shown to increase aggression, impair visuoconstructional abilities, and induce cognitive dysfunction—factors that collectively elevate the risk of automobile crashes [29,30].

Initial studies during the pandemic consistently documented a reduction in car crash rates during the early stages of the pandemic [31]. These decreases were mainly observed in less severe collisions occurring during rush hour or in minor-moderate crashes that were not fatal but would require hospitalization [32–36]. Conversely, fatal crashes increased, largely attributed to speeding and reduced traffic volume [37,38].

Beyond speeding, factors such as reckless driving, inattentiveness, and aggression have also been implicated in the rise of severe crashes, although no direct links to SARS-CoV-2's effects on the central nervous system (CNS) have been proposed [28]. Further, studies in numerous states have repeated these findings, associating increases risk-taking behaviors such as failure to use a seatbelt with increases in fatality and severity of crashes. In the same sense, studies

have also demonstrated that SARS-CoV-2 vaccine hesitancy was associated car crashes and severity, indicating that inherent risk hesitancy and associated risk taking maybe exacerbated by SARS-CoV-2 [29,30,34,39].

Further complicating the picture, while the total number of crashes decreased, fatal crashes increased due to single-vehicle crashes, often attributed to risk-taking behavior [40]. Demonstrating the possible neurophysiological role of SARS-CoV-2 as the main driver of increases in severe and fatal crashes, studies have shown increased risk-taking behavior in recovered COVID-19 patients, possibly due to ongoing damage to the limbic system, mainly the amygdala [41–44]. While some researchers have argued that such risk taking maybe secondary to boredom and direct effects of the pandemic mitigation efforts, this explanation fails to account for global patterns observed across diverse cultural contexts and varying levels of pandemic restrictions [45]. The changes in car crash composition and risk-taking behavior were demonstrated to be a global phenomenon, strengthening our hypothesis that the CNS related changes owing to SARS-CoV-2 are to blame [46,47].

Moreover, although some states suspended on-the-road testing for teenage drivers during the early pandemic, the outcomes did not significantly deviate from the norm, as most states do not require behind-the-wheel training. While young and newly licensed drivers typically have higher crash rates, these incidents are usually non-fatal [48–50]. Lower socioeconomic status (SES) remains a significant factor in road safety and risk-taking behavior within this demographic, reinforcing the link between SES and increased risk. Internationally, similar changes in driving behavior have been observed, such as in Greece and Saudi Arabia, where speeding and fatal crash rates increased despite robust pandemic mitigation efforts, further challenging the notion that these trends result solely from mitigation measures like stay-at-home orders [51,52]. The demographic and psychological profiles of individuals who defied pandemic mitigation measures may offer additional insights into these patterns. Such individuals—often characterized by lower SES, educational attainment, employment in the service industry, ownership of older or less safe vehicles, younger age, male gender, and a history of substance abuse—were more likely to be on the road during stay-at-home orders, increasing their exposure to SARS-CoV-2 and, consequently, their likelihood of subsequent reinfections [53–56]. Supporting this perspective, data suggest that drivers self-reported engaging in fewer risky behaviors during the pandemic [57]. However, international data, such as from Thailand, where a second stay-at-home order following a surge in COVID-19 cases resulted in an increase in crashes and fatalities, contradicts the notion that mitigation efforts alone can account for these changes [58] Additionally, the rise in motorcycle crashes and non-compliance with safety measures globally suggests a broader shift in risk-taking behavior, potentially linked to the neurophysiological effects of SARS-CoV-2 [59].

## Materials & methods

### Study design

In response to the growing SARS-CoV-2 and Long COVID-19 epidemic, we conducted an in-depth analysis of the correlation between acute COVID-19 cases and the incidence of car crashes across seven states. This analysis was stratified by SARS-CoV-2 vaccination rates, Long COVID-19 prevalence as reported by the Centers for Disease Control and Prevention (CDC), and early pandemic policies, specifically in states with minimal or no COVID-19 restrictions [60].

### Data collection

Data collection spanned March to May 2024, covering data from 2020 to 2023 to ensure finalization and completeness. To ensure robust findings, we amalgamated car crash data

from each state's Department of Transportation (DOT) or equivalent, cross-referencing with the National Highway Traffic Safety Administration (NHTSA) data repository to ensure accuracy and consistency. Specifically, crash data were obtained from the following sources: the TxDOT Crash Query Tool (C.R.I.S. Query) for Texas, the MassDOT IMPACT tool for Massachusetts, the Washington State Collision Analysis Tool for Washington State, the Iowa DOT ICAT for Iowa, the Connecticut Crash Data Repository for Connecticut, the Florida Highway Safety and Motor Vehicles (FLHSMV) Crash Dashboard for Florida, and the GDOT Crash Data Portal for Georgia [61–68]. This multi-source approach ensured that discrepancies or incomplete data from one repository could be cross-verified and resolved.

COVID-19 vaccination data were sourced from the CDC's National Center for Immunization and Respiratory Diseases (NCIRD) [69]. Given the inherent variability in test reporting and the widespread use of at-home antigen testing—results of which are largely unreported—we employed PCR positivity rates obtained from the U.S. Department of Health & Human Services/ Centers for Disease Control and Prevention (CDC) as a surrogate marker for COVID-19 incidence [70]. PCR positivity rates were validated using time-series data from the Pandemic Mitigation Collaborative (PMC) and wastewater SARS-CoV-2 RNA levels from the CDC [1,70,71]. Refined wastewater metrics, accounting for population size, wastewater flow rates, and environmental conditions, showed strong positive correlations with PCR positivity rates ($R^2 > 0.92$, $p < 0.001$), with validation performed via regression analysis. These validations confirm PCR positivity as a reliable proxy for COVID-19 case numbers in line with previous literature [72] not only validating PCR positivity rates as a reliable measure of COVID-19 incidence but also underscoring the utility of such in epidemiological analyses, particularly when traditional testing data may be incomplete or biased.

## Statistical reasoning

The Poisson regression model was used to assess the relationship between acute COVID-19 incidence and car crashes. Fixed effects were applied to control for state-specific characteristics, as the analysis focused on state-level data. State-level fixed effects accounted for differences in policy implementation, population density, and road usage patterns. Overdispersion was evaluated using the Pearson dispersion statistic. The absence of significant overdispersion justified the use of the Poisson model over a Negative Binomial alternative.

The log of the population was included as an offset in the Poisson regression equation to account for differences in population size among states. The outcome variable, the count of car crashes, was modeled as a log-transformed variable. Covariates included vaccination rates, Long COVID-19 prevalence, and PCR positivity rates. Interaction terms between mitigation measures and vaccination rates were considered but ultimately excluded due to lack of statistical significance and lack of available enforcement/compliance data. Descriptive statistics for all datasets are detailed in Table 1.

## Correlation analysis and heat map

A correlation analysis was conducted to assess whether changes in one variable were associated with changes in another. Highly correlated variables (above 0.75) were excluded from the multivariable model to avoid collinearity. Pearson's correlation coefficient was used to quantify the strength and direction of associations. To visualize correlations, a heat map was constructed, categorizing correlations as high (above 0.75), medium (0.5–0.75), or low (below 0.5). The heat map revealed no significant correlations among predictor variables, validating the independence of variables used in the Poisson regression model (Fig 1). The heat map also informed variable selection by excluding predictors with potential collinearity, thereby improving the robustness of the regression model.

**Table 1. Descriptive Statistics.**

| Parameter | Car Crash Rate, Mean ± STD, (Range) | P Value |
|---|---|---|
| **OVERALL** | 20.9 ± 8.1, (6.3 - 42) | – |
| **YEAR** | | *0.0057* |
| 2020 | 18.3 ± 7.4, (6.3 – 38.1) | |
| 2021 | 21.9 ± 8.4, (9.1 – 42.0) | |
| 2022 | 22.0 ± 8.0, (9.5 – 40.3) | |
| **STATE** | | *<.0001* |
| CT | 28.5 ± 5.0, (12.5 – 36.6) | |
| FL | 16.5 ± 1.9, (9.1 – 18.7) | |
| GA | 33.9 ± 4.9, (16.9 – 42.0) | |
| IA | 11.4 ± 2.0, (6.3 – 15.9) | |
| MA | 19.3 ± 3.5, (8.7 – 24.9) | |
| TX | 21.9 ± 2.6, (12.8 – 25.9) | |
| WA | 14.6 ± 2.3, (7.4 – 19.0) | |

Range was defined as minimum through maximum.

*P* values indicated the statistically significant difference at 95% confidence interval, which was calculated from ANOVA test.

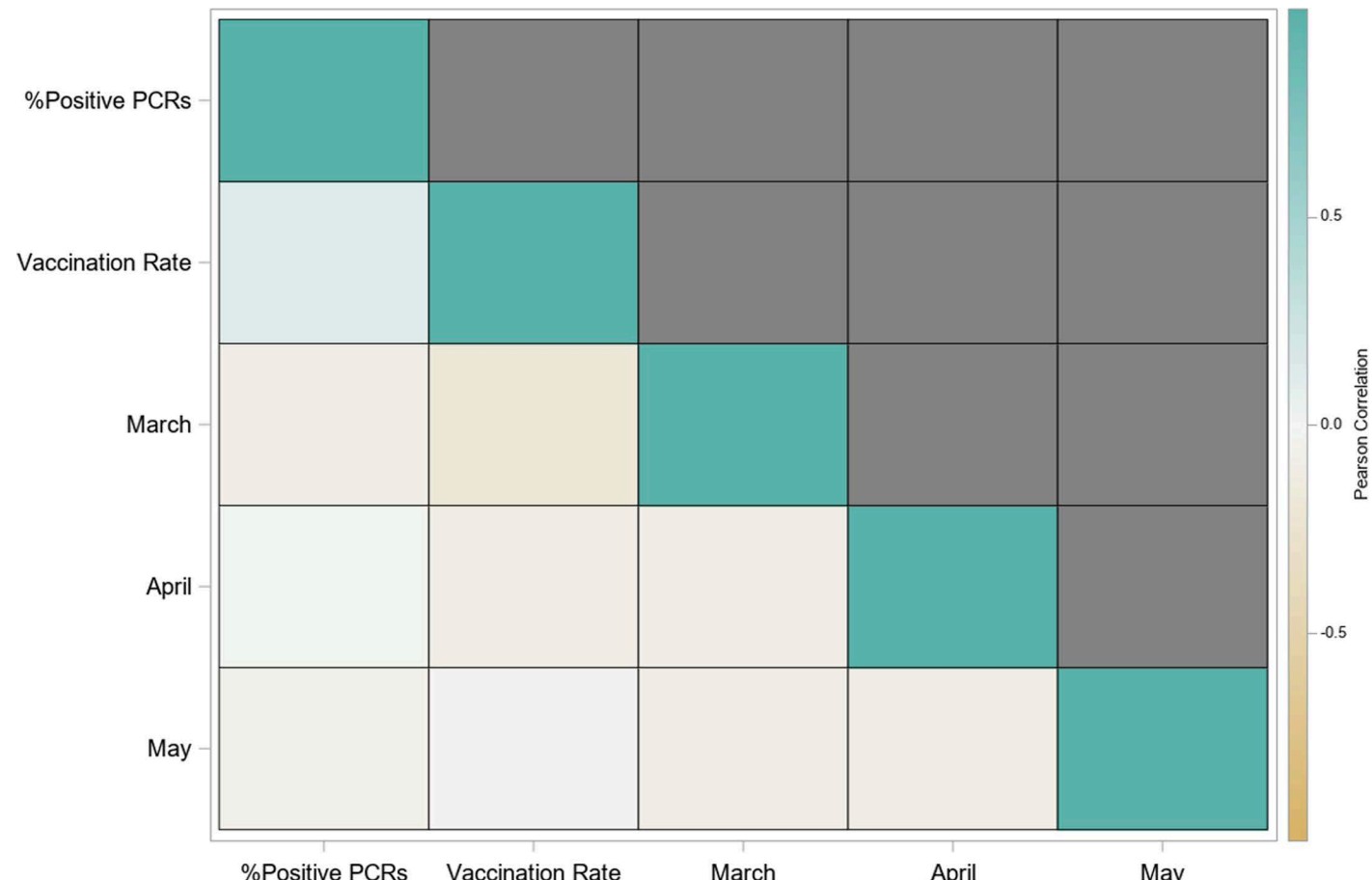

**Fig 1. Correlation Heat Map.**

## Software

All analyses were performed using SAS version 9.4 (SAS Institute Inc., Cary, NC). No custom scripts or extensions were applied to the software during the analysis.

## Supplementary data

The complete dataset is available in S1 and S2 Data. S1 Data (Wastewater Data) includes three tabs: percentage subsets of wastewater metrics, concentration data, and a master list of facilities. S2 Data provides the remaining datasets. Changes in facility names, primarily within the Verily-sourced wastewater data, are noted.

## Statistical analysis

The 2020–2022 data were fitted to a Poisson regression model. The log of population (per the US Census) was an offset variable for differences among seven states (Connecticut, Florida, Georgia, Iowa, Massachusetts, Texas, and Washington). (Table 2) Binary variables for March, April, and May were included separately because those were the months of the shutdowns in most US states and were included in the model that found no correlation between car crashes and concurrent COVID-19 rate (% positive PCR). The form of the equation with log-transformation was: $\log(\mu_i) = \beta_0 + \beta_1$ (COVID-19 rate) $+ \beta_2$ (vaccination rate) $+ \beta_3$ (March) $+ \beta_4$ (April) $+ \beta_5$ (May) $+ \log(\text{population}_i)$, Where: $\mu_i$ represents the expected number of car crashes in state i; $\log(\text{population}_i)$ is included as an offset to account for population differences.

A Poisson regression model was chosen for its effectiveness in modeling count data, particularly for rare events like car crashes, where the data is characterized by rare occurrences and skewed distributions, as supported by prior literature [25,37]. Fixed effects were used in the model to account for specific state-level differences, given the focus on evaluating how various state policies, mitigation efforts, and vaccination rates influenced crash rates to control for observed, non-random variability across states while ensuring the model's estimates were not biased by unobserved heterogeneity. This approach was appropriate given the study's focus on state-specific impacts of SARS-CoV-2 incidence and policy rather than broader variation across states, which would have warranted random effects. The log of the population was included as an offset variable to adjust for differing population sizes across states. Overdispersion, a common issue in count data where variance exceeds the mean, was assessed using the dispersion statistic (ratio of the deviance to the degrees of freedom), being critical in ensuring the model's robustness and accuracy, as overdispersion can lead to underestimation of standard errors, resulting in misleading inferences. In our model, overdispersion was specifically assessed by comparing the variance and mean of the outcome variable, with the dispersion statistic (ratio of the deviance to the degrees of freedom) suggesting minimal overdispersion (value close to 1), indicating that the Poisson model appropriately fits the data without underestimating standard errors. Consequently, a negative binomial model, typically used

Table 2. State Characteristics.

| | Connecticut (CT) | Florida (FL) | Georgia (GA) | Iowa (IA) | Massachusetts (MA) | Texas (TX) | Washington (WA) |
|---|---|---|---|---|---|---|---|
| **Long Covid Rate Per CDC (early 2024) (60)** | 5.2% | 5.9% | 6.2% | 7.1% | 5.3% | 6.5% | 5.2% |
| **Timing of Stay-at-Home Orders** | 03/23/20-05/20/20 | 04/02/20-05/04/20 | 04/02/20-04/30/20 | None | 03/24/20-05/18/20 | 04/02/20-04/30/20 | 03/24/20-05/31/20 |
| **Duration of Mask Mandates** | 683 Days | None | None | 82 Days | 388 Days | 250 Days | 625 Days |
| **Cumulative Adult (18+) Initial COVID-19 Vaccination Completion Rate on 12/30/21** | 74.6% | 63.4% | 51% | 59% | 74.6% | 57.7% | 68.6% |

for significant overdispersion, was deemed unnecessary due to the observed minor overdispersion. Ultimately, Poisson regression is well-suited to situations requiring adjustments for varying population sizes across different States, thus allowing for a more nuanced analysis of the incidence rate of car crashes as a function of predictor variables. By incorporating variables such as COVID-19 infection rates alongside other relevant covariates, Poisson regression has in this context facilitated a robust framework for analyzing the relationship between SARS-CoV-2 and road safety outcomes while accounting for state-level population differences and other contextual factors.

## Results

The increases in crash frequency temporally coincided with the initial dip in SARS-CoV-2 rates thanks to stay-at-home orders, and subsequently ever increased following removal of most mitigations and advent of ever-virulent variants, e.g., Delta and Omicron. Not surprisingly, our data demonstrate lower COVID-19 rates in states with more stringent mitigation efforts, namely masking mandates and higher vaccination rates. (Tables 2 and 3) As discussed above, to evaluate the relationship between variables, the correlation analysis conducted visually summarizes the correlations in a heat map (Fig 1), categorizing them as high (>0.75), medium (0.5–0.75), or low (<0.5). The heat map (Fig 1) revealed no significant correlations among predictor variables, validating their independence and inclusion in the Poisson regression model. Ultimately, our analysis demonstrated that acute COVID as measured by PCR positivity was associated with an odds ratio (OR) of 1.25 (1.23-1.26 95%CI) for subsequent car crashes indicating a 25% increase in crash risk associated with acute COVID-19. (Table 3). The results were consistent across univariate and multivariate analyses, demonstrating statistically significant results (p<0.05) in all states except Iowa and Washington. (Table 3)

**Table 3. Univariable and Multivariable Analysis on Car Crashes by COVID-19 Rate through Poisson Regression Model.**

| PARAMETER | OR (95% CI) | P Value[a] |
|---|---|---|
| **Univariate Model** | | |
| % Positive PCRs | 1.25 [1.23 - 1.26] | <.0001 |
| **Multivariate Model** | | |
| % Positive PCRs | 1.44 [1.42 - 1.47] | <.0001 |
| Vaccination | 0.98 [0.97 - 0.98] | <.0001 |
| March | 1.04 [1.04 - 1.05] | <.0001 |
| April | 1.05 [1.05 - 1.06] | <.0001 |
| May | 0.99 [0.99 - 0.99] | <.0001 |
| **Stratified by State** | | |
| STATE | OR (95% CI) | P Value[a] |
| CT | 3.19 [2.99 - 3.41] | **<.0001** |
| FL | 1.37 [1.33 - 1.42] | **<.0001** |
| GA | 1.65 [1.60 - 1.70] | **<.0001** |
| IA | 1.10 [0.99 - 1.22] | 0.0687 |
| MA | 5.22 [4.88 - 5.58] | **<.0001** |
| TX | 1.30 [1.27 - 1.32] | **<.0001** |
| WA | 1.02 [0.95 - 1.10] | 0.5608 |

[a]Bolded P value indicated statistical significance within a 95% confidence interval, i.e., P<0.05.

## Discussion

Previous studies utilizing similar Poisson regression has demonstrated a connection between the number of accumulated COVID-19 cases and an increase in road fatalities, but not simultaneously and in a temporally associated manner [73]. Our analysis extends these findings by establishing a temporal link, showing that car crashes are associated with increased COVID-19 rates acutely. Furthermore, we found that this increase in crashes is independent of Long COVID rates, focusing on acute infections as the primary driver. Interestingly, vaccination does not appear to confer a protective effect against crash risk, contradicting prior studies associating vaccine hesitancy with elevated crash risk [39]. States with extended mask mandates, like Connecticut, experienced prolonged periods of reduced traffic crash rates, whereas our inability to establish statistical significance in Iowa and Washington likely stems from data reporting issues and differences in pandemic-related behavioral changes, such as work-from-home arrangements in Washington and a lack of mitigation efforts in Iowa.

Further, we chose not to include gas prices as a variable, as previous data has demonstrated that gas demand is inelastic, meaning driving choices do not immediately reflect pricing, and this variable was deemed irrelevant to our focus on acute COVID-19. Additionally, over 10% of US vehicles currently on the road are powered by electricity reducing the relevance of traditional fuel price metrics [74,75].

Our observed odds ratio 1.25 [1.23-1.26] is comparable to that associated with a blood alcohol concentration (BAC) of 0.08%, the legal threshold for driving under the influence (DUI) in many states. Comparatively, states such as Utah, which set lower BAC limits of 0.05%, show similar crash risk odds ratios, underlining the substantial impact of acute COVID-19 on driving safety [76]. This finding is also consistent with odds ratios linked to various problematic driving behaviors, such as habitual speeding or running red lights—activities that are illegal and heavily enforced due to their significant risk profiles drawing the ire of public [77]. Although this study examined aggregate data cannot definitively determine the causality of COVID's effects on the central nervous system (CNS) & car crash frequency, the hypothesis is defensible both mechanistically and statistically.

Our analysis faced several limitations, including the rising prevalence of uninsured drivers, which may contribute to crash underreporting, and the inability to directly confirm acute COVID-19 infections in individuals involved in these crashes, given the population-level nature of the data. Despite these limitations, our findings provide substantial support for the hypothesis that acute COVID-19 contributes to increased car crash risk.

Some prior studies have reported decreases in car crashes frequencies concurrent with rising COVID-19 case rates. However, these studies were conducted in countries such as Japan and Greece, where stringent mitigation measures significantly restricted population mobility, leading to immediate reductions in traffic volume. Importantly, not all studies demonstrated corresponding decreases in fatalities beyond reductions in crash frequency [78,79]. While these studies addressed changes in crash occurrences alongside fatalities and injuries, our analysis highlights the distinct role of acute COVID-19 in contributing to the risk of car crashes. This distinction is critical, as crash severity involves unique risk factors that differ from those influencing crash frequencies alone. For example, prior research shows that population density and traffic density as inversely correlated risk factors for crashes and fatalities [80,81].

Previous studies on exploration of crash heterogeneity further support the importance of distinguishing between outcomes such as crash frequency and severity. The causal mechanisms underlying crashes of varying severity differ significantly, and generalizing across these categories can lead to erroneous conclusions, highlighting the need for targeted analysis [82]. Ultimately, this perspective strengthens our hypothesis: in settings without restrictive mobility

measures, the persistence of crashes suggests that acute COVID-19 may exacerbate risk through mechanisms such as impaired cognition or judgment.

To further elucidate the implications of COVID-19 sequelae on car crashes, additional research—such as case-control or prospective cohort studies—is urgently needed. This is particularly critical given the impacts of COVID-19 on the CNS, especially its effects on cognition, which may lead to judgment errors in high-stakes environments, such as flight decks or nuclear energy plants, with potentially catastrophic consequences.

## Conclusions

With the COVID-19 Public Health Emergency (PHE) expiring on May 11, 2023, society has primarily suspended protection against COVID-19. In light of lack of mitigations surrounding COVID-19 spread and the failure of the vaccination-only public health approach that has focused mainly on acute hospitalizations and deaths, further research is urgently needed into Long COVID as well as treatments to help manage the sequelae of an Acute Covid infection.

Legislatures and public health experts should not only view COVID-19 in the sense of acute mortality and morbidity. As brief neuropsychological tests are predictive of Long COVID and validated for driving risk, agencies responsible for driving licenses should implement a short questionnaire at license renewal inquiring about Long COVID/COVID and refer applicants to neuropsychological testing as needed. Perhaps even asking if drivers have had ongoing taste and smell disturbances post-COVID might link applicants with ongoing neurological sequelae of COVID [83–85].

Finally, clinicians, particularly those dealing with Long Covid patients in the cognitive setting, such as neurologists, must remember their obligation to report patients who potentially constitute medically impaired drivers. Such patients may include those with COVID-19 or those who suffer from the after-effects. Indeed, clinicians, particularly primary care practitioners, should contemplate warning patients with COVID-19 that they should minimize driving and report back if they feel any cognitive sequelae.

## Supporting information

**S1 Data. Dataset underlying the main analysis with tabs separating data for differing states.**
(XLSX)

**S2 Data. Wastewater data including three tabs: percentage subsets of wastewater metrics, concentration data, and a master list of facilities, changes in facility names, primarily within the Verily-sourced wastewater data, are noted.**
(XLSX)

## Acknowledgments

I would like to thank Kaitlyn Bruneau, LCSW with her assistance in data collection as well as Sara Anne Willette in her assistance with raw wastewater data collection and analysis.

## Author contributions

**Conceptualization:** Baran Erdik.

**Data curation:** Baran Erdik.

**Formal analysis:** Baran Erdik.

**Investigation:** Baran Erdik.

**Methodology:** Baran Erdik.

**Project administration:** Baran Erdik.

**Resources:** Baran Erdik.

**Supervision:** Baran Erdik.

**Validation:** Baran Erdik.

**Visualization:** Baran Erdik.

**Writing – original draft:** Baran Erdik.

**Writing – review & editing:** Baran Erdik.

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
