## [Decision Letter · Decision Letter 0]

26 Jun 2024

PGPH-D-24-01012

Driving under viral impairment: Linking acute SARS-CoV-2 infections to elevated car crash risks

Dear Dr. Erdik,

Thank you for submitting your manuscript to PLOS Global Public Health. After careful consideration, we feel that it has merit but does not fully meet PLOS Global Public Health’s publication criteria as it currently stands. Therefore, we invite you to submit a revised version of the manuscript that addresses the points raised during the review process.

Please note that we have only been able to secure a single reviewer to assess your manuscript. We are issuing a decision on your manuscript at this point to prevent further delays in the evaluation of your manuscript. Please be aware that the editor who handles your revised manuscript might find it necessary to invite additional reviewers to assess this work once the revised manuscript is submitted. However, we will aim to proceed on the basis of this single review if possible. 

The comments from the reviewer are below. I agree with the reviewer that a more thorough analysis of the existing literature on the associations between COVID 19 and car crashes is needed (although it is not a requirement that you cite the specific papers mentioned by the reviewer).

I also agree that a lot more methodological details are needed. At present it is not clear what data you used, when and how you obtained it, and what variables were included. Please provide more precise details of the datasets you accessed such that other researchers would be able to reproduce your research. Please also provide whatever data you can as supplementary files (see https://journals.plos.org/globalpublichealth/s/data-availability).

Please also provide details on your raw wastewater data collection and analysis, including sources of sample, lab procedures, and analytic techniques.

We look forward to receiving your revised manuscript.

Kind regards,

Steve Zimmerman, PhD

PLOS Staff Editor

Journal Requirements:

Additional Editor Comments (if provided):

Reviewers' comments:

Reviewer's Responses to Questions

**Comments to the Author**

1. Does this manuscript meet PLOS Global Public Health’s publication criteria ? Is the manuscript technically sound, and do the data support the conclusions? The manuscript must describe methodologically and ethically rigorous research with conclusions that are appropriately drawn based on the data presented.

Reviewer #1: Yes

2. Has the statistical analysis been performed appropriately and rigorously?

Reviewer #1: Yes

3. Have the authors made all data underlying the findings in their manuscript fully available (please refer to the Data Availability Statement at the start of the manuscript PDF file)?

Reviewer #1: Yes

4. Is the manuscript presented in an intelligible fashion and written in standard English?

Reviewer #1: Yes

5. Review Comments to the Author

Reviewer #1: This paper examines the association between COVID-19 and car crashes across U.S. states, correlating with the implemented mitigation strategies, vaccination rates, and long COVID prevalence. The following remarks should be taken into consideration in the revised version of this paper:

The authors should structure a more thorough literature review, extending the existing insights to reflect the extensive research performed since the COVID-19 spread. The focus should be more on the traffic impacts and the origins of traffic crashes, highlighting that the increase in crashes is a combination of behavioral changes and primarily the reduction in traffic volumes. See:

• Adanu, E. K., Brown, D., Jones, S., & Parrish, A. (2021). How did the COVID-19 pandemic affect road crashes and crash outcomes in Alabama?. Accident Analysis & Prevention, 163, 106428.

• Sekadakis, M., Katrakazas, C., Michelaraki, E., Kehagia, F., & Yannis, G. (2021). Analysis of the impact of COVID-19 on collisions, fatalities, and injuries using time series forecasting: The case of Greece. Accident Analysis & Prevention, 162, 106391.

It is significant, based on the literature review, to highlight the innovation of this study. Furthermore, the methodology should be extended to include the databases used, methods employed, and descriptive statistics of the datasets. Additionally, the authors should substantiate why Poisson regression was chosen. The discussion and conclusions should be strengthened in logical flow and discussed more thoroughly.

6. PLOS authors have the option to publish the peer review history of their article (what does this mean? ). If published, this will include your full peer review and any attached files.

**Do you want your identity to be public for this peer review?** For information about this choice, including consent withdrawal, please see our Privacy Policy .

Reviewer #1: No

---

## [Decision Letter · Decision Letter 1]

22 Jan 2025

PGPH-D-24-01012R1

Driving under viral impairment: Linking acute SARS-CoV-2 infections to elevated car crash risks

Dear Dr. Baran Erdik,

Thank you for submitting your manuscript to PLOS Global Public Health. After careful consideration, we feel that it has merit but does not fully meet PLOS Global Public Health’s publication criteria as it currently stands. Therefore, we invite you to submit a revised version of the manuscript that addresses the points raised during the review process.

Please address the following observations to enhance the clarity, consistency, and rigor of your work:

Abstract and Methods Consistency:

In the abstract, the analysis is described as covering the period from 2020 to 2023 (lines 19–20, page 2). However, in the Methods section (page 8, line 143), data collection is stated to have occurred between March and May 2024. Please clarify the exact data period used in the study.

Materials and Methods:

Specify the exact number of states included in the analysis instead of referring to “several” states.Elaborate on the extent to which crude and refined wastewater metrics were utilized.Include explicit references for the sources of information described in the Methods section to ensure transparency.

Results Section:

Ensure there is no repetition in the text. For example, the wording in lines 181–185 (page 10) is repeated.Correct the duplication of Table 2 to avoid confusion.

Terminology:

Avoid using the term “accidents” and consistently use “crashes” throughout the manuscript.

Methods Section:

Provide a detailed description of the Poisson model used, including whether fixed or random effects were applied, and justify the choice in light of state-level analysis.Describe how overdispersion was assessed and explain why a Negative Binomial model was not used.Move the description of the heat map from the Results section to the Methods section and clarify how it was constructed and used to identify variables for the models.Revise the equation to properly include the log of the population as an offset, and ensure the outcome is clearly described as a log-transformed variable.State the software used to perform the analyses.

Discussion Section:

Revise the statement in line 269 (page 15) to avoid conflating fatalities and crashes, as these are distinct outcomes. Ensure the discussion aligns with the specific outcome being analyzed.

References:

Include and properly cite the following reference, which was mentioned in your discussion:

R.R. Knipling, “Crash Heterogeneity: Implications for Naturalistic Driving Studies and for Understanding Crash Risks,” Transp. Res. Rec., 2663 (2017), pp. 117-125, 10.3141/2663-15.

We look forward to receiving your revised manuscript.

Kind regards,

Jose Ignacio Nazif-Munoz, Ph.D.

Academic Editor

Journal Requirements:

1. If the authors have adequately addressed your comments raised in a previous round of review and you feel that this manuscript is now acceptable for publication, you may indicate that here to bypass the “Comments to the Author” section, enter your conflict of interest statement in the “Confidential to Editor” section, and submit your "Accept" recommendation.

Reviewer #1: All comments have been addressed

2. Does this manuscript meet PLOS Global Public Health’s publication criteria ? Is the manuscript technically sound, and do the data support the conclusions? The manuscript must describe methodologically and ethically rigorous research with conclusions that are appropriately drawn based on the data presented.

Reviewer #1: Yes

3. Has the statistical analysis been performed appropriately and rigorously?

Reviewer #1: Yes

4. Have the authors made all data underlying the findings in their manuscript fully available (please refer to the Data Availability Statement at the start of the manuscript PDF file)?

Reviewer #1: Yes

5. Is the manuscript presented in an intelligible fashion and written in standard English?

Reviewer #1: Yes

6. Review Comments to the Author

Reviewer #1: The authors have thoroughly revised the manuscript in response to the reviewer's comments. The revised version has been improved and now meets the journal's publication requirements.

7. PLOS authors have the option to publish the peer review history of their article (what does this mean? ). If published, this will include your full peer review and any attached files.

**Do you want your identity to be public for this peer review?** For information about this choice, including consent withdrawal, please see our Privacy Policy .

Reviewer #1: No

---

## [Editor Report · Decision Letter 2]

25 Feb 2025

Driving under viral impairment: Linking acute SARS-CoV-2 infections to elevated car crash risks

PGPH-D-24-01012R2

Dear Dr. Baran Erdik,

We are pleased to inform you that your manuscript 'Driving under viral impairment: Linking acute SARS-CoV-2 infections to elevated car crash risks' has been provisionally accepted for publication in PLOS Global Public Health.

Best regards,

Jose Ignacio Nazif-Munoz, Ph.D.

Academic Editor